# FEW-SHOT LEARNING WITH REPRESENTATIVE GLOBAL PROTOTYPE

## ABSTRACT

Few-shot learning is often challenged by low generalization performance due to the assumption that the data distribution of novel classes and base classes is similar while the model is trained only on the base classes. To mitigate the above issues, we propose a few-shot learning with representative global prototype method. Specifically, to enhance the generalization to novel classes, we propose a method to jointly train the base classes and the novel classes, using selected representative and non-representative samples to optimize representative global prototypes, respectively. Additionally, a method that organically combines the sample of base classes conditional on semantic embedding to generate new samples of novel classes with the original data is proposed to enhance the data of novel classes. Results show that this training method improves the model's ability to describe novel classes, improving the classification performance for a few shots. Intensive experiments have been conducted on two popular benchmark datasets, and the experimental results show that this method significantly improves the classification ability of few-shot learning tasks and achieves state-of-the-art performance.

## 1 INTRODUCTION

The advantages of deep learning depend on big data, and machines can learn effectively under the drive of big data. However, an insufficient amount of data leads to problems, such as model overfitting, which makes the model not fit well on the dataset outside the training data, showing weak generalization ability Kang et al. (2019). Thus, few-shot learning for small datasets has become a key technology to solve this type of problem. Few-shot learning (FSL) refers to the use of a small amount of labeled data per class training to make the model show higher performance, in line with the law of human learning.

Meta-learning is the basis of most of the existing FSL methods. In meta-learning, training is modeled after a test, i.e., both base and novel classes learn a task consisting of a few samples from N-way K-shot, after which knowledge is transferred from the base classes in the form of appropriate initial conditions Finn et al. (2017); Fei et al. (2020); Zhang et al. (2020), embeddings Vinyals et al. (2016); Chopra et al. (2005); Hadsell et al. (2006); Sung et al. (2018) or optimization strategies Ravi & Larochelle (2017). There is, however, a fundamental limitation to all of these approaches: the models assumed that the distribution of base class data and novel class data was similar, so they mostly learned using only base class data, which does not guarantee that they are generalizable to novel class data.

A few-shot learning method uses the transferability of knowledge to learn the ability to characterize samples from novel classes with only a few markers, while traditional machine learning and deep learning feed all classes of samples to the model at the beginning of training. Once the data distribution has low consistency, transferred knowledge is not applicable to generalize the novel class of data. Thus, learning the global representation of all class data together can alleviate the problem of model overfitting on the base classes like machine learning and deep learning. The novel class data is learned at the beginning, and the metric novel class data and the global representation strengthen the discriminative nature of the novel class samples Wang et al. (2018); Li et al. (2019). At the same time, the sample synthesis strategy of Wang et al. (2018) is used to enhance the samples of the novel classes to alleviate the sample imbalance. However, since the global prototype is optimized directly using all samples, it is not representative. Moreover, the sample generation method actually requires

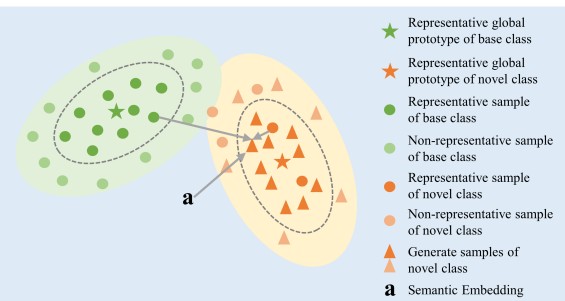

Figure 1: An illustration of representative global prototype. Combining representative samples from the base class, semantic embeddings, and novel class samples allows us to generate new samples, which are then used to train a representative global prototype.

that the base classes data and the novel classes data are similarly distributed, and do not break the fundamental restrictions Wang et al. (2018). Finally, the generation process is not directly involved in sample training and cannot optimize the quality of the generated samples.

Therefore, we propose a new FSL method to solve the problem of poor generalization ability for few-shot data and sample imbalance to better describe the data of the novel class. The novel class and the selected representative and non-representative data from the base class are fed together into the network to jointly learn the representational ability of the data, and thus it is called a representative global prototype. In our view, the FSL model does not have to meet the barrier of similar distribution of the novel and base class data, and is better suited to identify novel classes when it is trained with novel class data.

However, trying to learn the global prototype of the base and novel class requires overcoming the problem of data imbalance with sparse data in the novel class. To overcome the strong correlation between the base and novel class data distribution, we propose a new sample generation strategy. Specifically, as in Xu & Le (2022), more representative samples are selected to generate more representative features for the novel class using the CVAE Sohn et al. (2015) generative model conditional on semantic embedding. In the above operation, we obtain enhanced data strongly correlated with the novel class, with the base class as auxiliary information.

Our main contributions are summarized as follows: (1) We propose a novel jointly training strategy for few-shot learning via representative and non-representative samples to break the assumption of meta-learning. (2) We propose a sample synthesis method to enhance novel classes of data. (3) Experiments show that our method exhibits state-of-the-art performance on both miniImageNet and tieredImageNet datasets.

## 2 RELATED WORK

### 2.1 FEW-SHOT LEARNING

Few-shot learning comes into play when we only have very limited training samples. Most of the recent deep learning approaches rely on meta-learning or learning-to-learn Hochreiter et al. (2001) strategy, which improves the performance of the novel task by the provided data set and the meta knowledge extracted across tasks by a meta-learner. Specifically, the meta-learner transfers knowledge learned from many base class tasks to help few-shot learning to complete training tasks in novel classes.

Presently, representative FSL via meta-learning can be divided into three categories: fine-tune based, metric based, and optimization based. (1) The methods of the first category aim to learn suitable initialization parameters for training novel class samples in order to train a novel classifier faster and better Finn et al. (2017); Rusu et al. (2019); Nichol & Schulman (2018); Fei et al. (2020); Zhang et al. (2020). The models of the second category are learning a good metric, the core of which is to learn effective discriminative features through a kernel function. Specifically, metric learning calculates the similarity between two samples by learning an embedding module and a metric module. The

samples of all classes are embedded into the vector space through the embedding module, and then a similarity score is given based on the metric module Vinyals et al. (2016); Chopra et al. (2005); Hadsell et al. (2006); Sung et al. (2018); Snell et al. (2017). The methods of the third category adapt optimizers for meta-learning scenarios by replacing general optimization methods Munkhdalai & Yu (2017); Ravi & Larochelle (2017). However, the above meta-learning based methods are all limited by one barrier: the samples of the novel class do not appear in the training phase, so the model is prone to overfitting on the base class.

## 2.2 GLOBAL REPRESENTATION LEARNING

A model overfitting problem can be alleviated by learning the global representation of all classes data together Li et al. (2019); Xi et al. (2022). It aims to learn a global representation for the classification of novel classes samples. Nevertheless, not all labeled samples are equally important, and non-representative samples can influence the accuracy of global representations. Model training can involve different sampled instances depending on the sampling strategy. Most mainstream work focuses on samples that affect decision boundaries, which is more common in active learning, in which training samples affecting classification are selected using different uncertainty metric Bappy et al. (2017); Li & Guo (2013); Zhao et al. (2021). In contrast, the representativeness of the data distribution is of less concern when sampling. In FSL, an approach that uses only representative samples to fit global representations is proposed by Xu & Le (2022), but it ignores the overall distribution of the data. A representative global representation is therefore proposed. To optimize the representativeness of the global representation, two loss functions are generated using representative samples and non-representative samples.

## 2.3 CONDITIONAL VARIATIONAL AUTOENCODER

In the past, conditional VAE has been used to model feature distributions for a variety of computer vision tasks, including image classification Kim et al. (2019); Schönfeld et al. (2019); Xing et al. (2019); Zhang et al. (2019), image generation Esser & Sutter (2018); Liu et al. (2017), image recovery Du et al. (2020), or video processing Pan et al. (2019). Essentially, VAE models are trained to learn the data distribution of the training set. As reported in Xi et al. (2022); Wang et al. (2018); Li et al. (2019) that VAE can be used to generate data for novel classes based on training fixed parameters from base classes. However, the problem is that the parameters learned using only the base class data, which are suitable for the base class to generate better data, but not necessarily for the novel class data. Therefore, the method of using CVAE Sohn et al. (2015) was proposed to combine the features of the extracted base class data conditioned on semantic embedding to generate features or images of the novel class data Xu & Le (2022). In this way, the novel class data can be generated more accurately by using common semantic embeddings as a bridge between the novel class and the base class. However, these methods are essentially still under the assumption of meta-learning. Therefore, we propose a new sample generation strategy that makes full use of the novel and base class data to synthesize new data. This does not depend on the parameters of the base class data training but also facilitates the classification of the novel class.

## 3 METHOD

The key idea of our model is to learn representative global prototype with the base and novel class jointly using the representative and non-representative samples. Furthermore, we employ a new sample generation strategy for novel classes to overcome the sample imbalance problem. In this section, we will first discuss these two key components before reviewing the entire process.

## 3.1 PROBLEM DEFINITION

In this section, we assign data. There are a total of $N$ classes of samples, denoted as $C_{all} = \{c_1, c_2, \cdots, c_N\}$. The class set $C_{all}$ consists of two disjoint sets: a set of base classes $C_{base}$ and a set of novel classes $C_{novel}$. Please note, the classes of training set $D_{train}$ are from $C_{all}$ but not $C_{base}$. And a test set $D_{test}$ with samples from classes in $C_{novel}$. The sets of class $D_{train}$ and $D_{test}$ are disjoint, $i.e. D_{train} \cap D_{test} = \emptyset$. In $D_{train}$, the samples from the base class have enough labels but are very limited from the novel class, with only $n_{few}(n_{few} \leq 5)$ labeled samples. A total of $n_s$

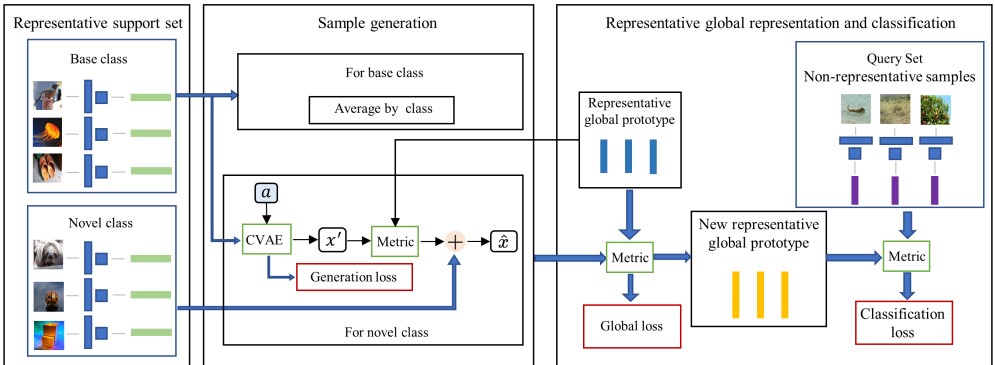

Figure 2: **Overview.** First, we propose a sample generation method to generate new samples of novel classes in the representative support set. Second, the representative global prototype are updated according to the similarity score of features of each class and representative global prototype, and the new representative global prototype and non-representative samples are then used to classify query images. The classification loss of query images, global loss, and generation loss are used to jointly optimize the global representations and the generation module.

representative base classes samples and original small amount of novel classes samples are sampled from $D_{train}$ to form a support set $S = \{(\mathbf{x}_i, y_i), i = 1, \cdots, n_s \times N\}$, and the non-representative samples are formed a query set $Q = \{(\mathbf{x}_k, y_k), k = 1, \cdots, n_q \times N\}$.

## 3.2 REPRESENTATIVE GLOBAL PROTOTYPE LEARNING

As a result of the lack of generalization ability for models trained only on base classes, a joint training strategy is proposed to learn a global prototype. Furthermore, to enhance its representativeness, only the representative support set is used to compute the global loss, while the classification loss is computed using the query set consisting of non-representative samples to compensate for the global prototype's incompleteness.

Specifically, we use representativeness as a criterion for selecting samples. $S = \{\mathbf{x}_j \mid p(\mathbf{x}_j \mid \mu_i, \Sigma_i) > \epsilon\}$ is a representative set of training samples of the base class selected by the method of Xu & Le (2022), where $p(\cdot)$ is the probability density of Gaussian distribution $\mathbf{x}_j$, $\mu_i$ is the average of all vectors in class $i$, the covariance matrix $\Sigma_i$ is the distribution of class $i$. A artificial threshold of $\epsilon$ is used to filter out samples with low probability. Identifies the sample easier if the sample is close to the mean vector $u_i$, which is its probability $p(\cdot)$ is large. For training the global loss, we selected the support set that is closest to the center vector.

A global loss is calculated by metricating the visual features of samples and all global representations of each class $\mathbf{G} = \{\mathbf{g}(i), i \subseteq C_{all}\}$, where $\mathbf{g}(i)$ is the global prototypical representation of the class $i$. The initialization of $\mathbf{g}(i)$ is the mean of embedding features of all the selected representative samples in $S$. For each visual feature $x_j$, the similarity vector $\mathbf{V}_j = \left[v_j^1, v_j^2, \cdots v_j^N\right]^T$ is obtained by a metric kernel function, where the $i$-th element is the similarity score of class $i$ in the embedding space:

$$v_j^i = - \|\mathbf{x}_j - \mathbf{g}(i)\|_2 \tag{1}$$

In this paper, in order to bring the global prototype closer to a representative sample, i.e.: larger posterior probability and more identifiable sample, we define a loss for class $i$ in the representative support set (called global loss).

$$\mathcal{L}_g = CE(y_j, \mathbf{V}_j) \tag{2}$$

where $CE(\cdot)$ is a cross entropy loss.

Immediately afterward, we utilize the representative global prototype as a class representative and identify non-representative samples of the query set. Due to the large number of non-representative samples and the fact that we consider the most uncertain samples to be more beneficial for the classification task Settles (2009). This reduces the training cost and speeds up the training process by filtering out some samples. Inspired by the largest margin uncertainty Joshi et al. (2009), we measure the maximum and minimum probability values if a sample belongs to a certain class. Obviously, if the difference between the two is significantly larger, then the classifier has a higher degree of confidence that the sample belongs to the class. Alternatively, if the difference is small, the sample is in the tail of a Gaussian distribution and its class is harder to identify. Therefore, we implement the criterion of largest margin uncertainty to choose the samples that are most difficult to identify class affiliation as our query set.

$$Q = \{\mathbf{x}_j \mid max\{p(y_i \mid \mathbf{x}_j)\} - min\{p(y_i \mid \mathbf{x}_j)\} < \beta\} \tag{3}$$

Once the query set data has been determined, we measure representative global prototypes and non-representative samples. In order to make the representative global prototype differentiable in backward propagation, softmax is used as the normalization function. At this point, the similarity vector $\mathbf{V}_j$ is normalized to a probability distribution $\mathbf{P}_j = \left[p_j^1, p_j^2, \cdots, p_j^N\right]^T$.

The global prototypical representation of each class is then updated according to:

$$\mathbf{g}_{new}(i) = \mathbf{G}\mathbf{P}_j \tag{4}$$

where $\mathbf{G}$ is consisting of the representative global prototype before being updated.

The similarity score $\mathbf{W}_k = \left[w_k^1, w_k^2, \cdots, w_k^N\right]^T$ between the new representative global prototype and the non-representative sample $\mathbf{x}_k$ in the query set $Q$ is defined as:

$$w_k^i = -\|\mathbf{x}_k - \mathbf{g}_{new}(i)\|_2 \tag{5}$$

A loss for the non-representative sample $\mathbf{x}_k$ in the query dataset $\mathbf{Q}$ (called classification loss) is defined as:

$$\mathcal{L}_{cla} = CE(y_k, \mathbf{W}_k) \tag{6}$$

The representative global prototype is jointly optimized by joint training of samples from the base class and the novel class by $\mathcal{L}_g$ and $\mathcal{L}_{cla}$, taking into account both the sample distribution and the classification performance. In contrast to methods using only representative samples, our approach considers the completeness of the data distribution. With a simple sample selection strategy, we reduce the amount of training data in comparison to an all-sample approach. The next step is to optimize a representative global prototype by jointly generating loss.

### 3.3 FEATURE GENERATION WITH CVAE

We propose a sample generation strategy to generate samples of the novel class without the assumption of meta-learning to solve the sample imbalance problem caused by limited data (few shots). Two steps are involved in the generation of features in this paper: (1) The features of the novel class are generated from original samples of the base class and semantic attributes. (2) All the features obtained in the first step and the original features of the novel class are used to synthesize new features.

Specifically, we combine the base class data and semantic embedding to generate new features by CVAE model. The selected representative samples are fed into the CVAE model to generate more representative features conditioned on the semantic embedding, the generate loss function is defined as:

$$\begin{aligned} \mathcal{L}_{CVAE} = \sum_{i=1}^{C_{base}} \sum_{\mathbf{x} \subseteq S} KL(q(\mathbf{z} \mid \mathbf{x}, \mathbf{a}^i) \parallel p(\mathbf{z} \mid \mathbf{a}^i)) \\ - E_{q(\mathbf{z} \mid \mathbf{x}, \mathbf{a}^i)}(logp(\mathbf{x} \mid \mathbf{z}, \mathbf{a}^i)) \end{aligned} \tag{7}$$

where $\mathbf{a}^i$ is the semantic embedding of class $i$. And the semantic embeddings are natural language descriptions of classes that contain attribute vectors of class information. The first term is the regularization term, which aligns the variation term $q(\mathbf{z} \mid \mathbf{x}, \mathbf{a})$ to the prior distribution $p(\mathbf{z} \mid \mathbf{a})$ through the Kullback-Leibler divergence. The second term is the reconstruction loss, which aims to make the features $\mathbf{z}$ from the encoder generated by the decoder $p(\mathbf{x} \mid \mathbf{z}, \mathbf{a})$ approximate the original input features. And it is stipulated that $N(0, I)$ is the prior distribution.

Regarding the method of generating features $\mathbf{x}'_j$ of novel classes, we input the semantic embedding $\mathbf{a}^i$ and latent variable $\mathbf{z}$ of the class in each novel class into the decoder to obtain new features of the corresponding class.

Using the method on the original base class samples, we will obtain a total of $k_t$ samples for every novel class. After that, we further synthesize new features from $k_t$ samples of each class. To enhance the distinguish ability of the generated features $\mathbf{x}'_j$, the similarity between the features $\mathbf{x}'_j$ and the global representation $\mathbf{g}(i)$ is computed, denoted as: $\mathbf{U}_j = \left[ u^1_j, u^2_j, \cdots, u^t_j \right]^T$.

$$u^i_j = - \left\| \mathbf{x}'_j - \mathbf{g}(i) \right\|_2 \tag{8}$$

Please note that only the score with the highest similarity to the novel classes are selected here.

To obtain the final enhanced feature $\hat{\mathbf{x}}_j$, normalize the similarity score $\mathbf{U}_j$ using softmax to $\mathbf{P}_{novel} = \left[ p^1_j, p^2_j, \cdots, p^t_j \right]^T$, and add it as the weight of generating features $\mathbf{x}'_j$ with the original global prototype representation.

$$\hat{\mathbf{x}}_j = \mathbf{g}(i) + \mathbf{x}'_j \mathbf{P}_{novel}, i \subseteq C_{novel} \tag{9}$$

By the above operation, we obtain the enhanced data strongly related to the novel classes with the base classes as auxiliary information. Finally, the $\hat{\mathbf{x}}_j$ and $\mathbf{x}_j$ in the novel class and the base class are fed to the model to optimize the representative global prototype.

## 3.4 MODEL REVIEW

We divide the whole model into two parts as Figure 2: representative global prototype learning and sample generation. The base class, the novel class and the semantic embedding are utilized to jointly generate the data of the novel class. After that, the novel and base class data are jointly trained to optimize the global representation. As an additional step, the representative samples are trained in order to optimize the representative global prototype. The loss of the entire model is as follows:

$$\mathcal{L}_{total} = \mathcal{L}_g + \mathcal{L}_{cla} + \mathcal{L}_{CVAE} \tag{10}$$

During the test phase, the same procedure is used to predict labels for unlabeled data in $C_{novel}$. To increase the number of samples in the novel class, a sample generation strategy is used. Our next step is to compute the Euclidean distance between the feature vectors of the test samples and the representative global prototype in the novel class in order to conduct the nearest neighbor search.

## 4 EXPERIMENTS

In this section, we evaluate and compare our method with state-of-the-art approaches on two datasets, i.e., miniImageNet and tieredImageNet. As a first step, we explain the detail settings. Then, we compare the proposed method with several state-of-the-art few-shot approaches to prove how effective it is. The experimental results are analyzed at the end of this section.

## 4.1 DATASETS

**miniImageNet Dataset** Vinyals et al. (2016) is a lightweight subset of the large-scale dataset ImageNet. It contains a total of 60,000 color images with an image size of $84 \times 84$. The set of these images is labeled with a size of 100, so there are 600 classes in each class. The data is divided into

Table 1: Few-shot classification accuracies (%) with 5-way on miniImageNet and tieredImageNet.

| model | backbone | miniImageNet | | tieredImageNet | |
|---|---|---|---|---|---|
| | | 1-shot | 5-shot | 1-shot | 5-shot |
| Matching NetVinyals et al. (2016) | ResNet-12 | 65.64±0.20 | 78.72±0.15 | 68.50±0.92 | 80.60±0.71 |
| MAMLFinn et al. (2017) | ResNet-18 | 64.06±0.18 | 80.58±0.12 | - | - |
| SimpleShotWang et al. (2019) | ResNet-18 | 62.85±0.20 | 80.02±0.14 | 69.09±0.22 | 84.58±0.16 |
| CANHou et al. (2019) | ResNet-12 | 63.85±0.48 | 79.44±0.34 | 69.89±0.51 | 84.23±0.37 |
| S2M2Mangla et al. (2020) | ResNet-12 | 64.06±0.18 | 80.58±0.12 | - | - |
| TADAMOreshkin et al. (2018) | ResNet-12 | 58.50±0.30 | 76.70±0.30 | 62.13±0.31 | 81.92±0.30 |
| AM3Xing et al. (2019) | ResNet-12 | 65.30±0.49 | 78.10±0.36 | 66.22±0.75 | 82.79±0.48 |
| Variational FSLZhang et al. (2019) | ResNet-12 | 61.23±0.26 | 77.69±0.17 | - | - |
| MetaOptNetLee et al. (2019) | ResNet-12 | 62.64±0.61 | 78.63±0.46 | 65.99±0.72 | 81.56±0.53 |
| Robust20-distillDvornik et al. (2019) | ResNet-18 | 63.06±0.61 | 80.63±0.42 | 65.43±0.21 | 70.44±0.32 |
| MELR Fei et al. (2020) | ResNet-12 | 67.40±0.43 | 83.40±0.28 | 72.14±0.51 | 87.01±0.35 |
| RFS Tian et al. (2020) | ResNet-12 | 62.02±0.63 | 79.64±0.44 | 69.74±0.72 | 84.41±0.55 |
| Neg-Cosine Liu et al. (2020a) | ResNet-12 | 63.85±0.81 | 81.57±0.56 | - | - |
| IEPTZhang et al. (2020) | ResNet-12 | 67.05±0.44 | 82.90±0.30 | 72.24±0.50 | 86.73±0.34 |
| Meta-BaselineChen et al. (2021) | ResNet-12 | 63.17±0.23 | 79.26±0.17 | 68.62±0.27 | 83.29±0.18 |
| ProtoNetYe et al. (2020) | ResNet-12 | 62.39 | 80.53 | 68.23 | 84.03 |
| E3BMLiu et al. (2020b) | ResNet-12 | 64.09±0.37 | 80.29±0.25 | 71.34±0.41 | 85.82±0.29 |
| R-SVAEXu & Le (2022) | ResNet-12 | 74.84±0.23 | 83.28±0.40 | 76.98±0.65 | 85.77±0.50 |
| GP(Ours) | ResNet-12 | 75.92±0.08 | 84.03±0.53 | 79.85±0.27 | 87.03±0.10 |
| RGP w/o (Ours) | ResNet-12 | 76.15±0.27 | 84.28±0.35 | 80.01±0.36 | 87.09±0.42 |
| RGP (Ours) | ResNet-12 | **76.41±0.24** | **84.52±0.31** | **80.33±0.17** | **87.18±0.36** |

three parts by labels. The training set contains 64 classes, the validation set contains 16 classes, and the remaining 20 classes are used as the test set.

**tieredImageNet Dataset** Ren et al. (2018) is also a subset of ImageNet like miniImageNet. ImageNet categories are sampled and grouped according to a hierarchical structure to separate the data into broader classes, and the image size is also 84 × 84. A total of 608 classes are contained in tieredImageNet, which has 34 categories containing between 10 and 30 classes. Among these, 20 categories / 351 classes make up the training set, 6 categories / 97 classes make up the validation set, and the remaining 8 categories / 160 classes make up the test set.

## 4.2 DETAIL SETTINGS

A feature extractor based on ResNet-12 with feature representations output from the final layer was trained using data from the base class before global training was conducted. A feature extractor is then used to average the visual features of all class samples to initialize the representative global prototype. Adam acts as an optimizer to train the entire network at $10^{-4}$. The dimensions of the latent space and semantic vector are both set to 512. Our semantic embeddings are extracted from word2vec Mikolov et al. (2013) trained using a skip-gram model on the Wikipedia corpus. We use the same feature extractor on images in the support and query sets. During our experiments, $\alpha$ is set to 0.9 and $\beta$ is set to 0.1.

## 4.3 RESULTS

We compare with state-of-the-art few-shot learning methods, among them, our method is most related to Xu & Le (2022). To be fair, all methods use feature extractors with a similar number of layers, namely: ResNet-12 and ResNet-18. In the remainder of the paper, we denote the model of training a global prototype using all data as **GP** and the training model with the selected representative and non-representative data as **RGP**, and denote the model only use representative data but no non-representative data as **RGP w/o**.

We show in Table 1 the classification results for the 5-way k-shot (k = 1 or 5) on the dataset. The results show that our method outperforms state-of-the-art methods, especially compared to R-SVAE which combines CLIP and utilizes representative samples. Furthermore, not only did the use of selected representative samples and non-representative partial samples not degrade the classification performance of the model, but the RGP showed a slight improvement in performance compared to the GP using full training. For example, the RGP outperformed the GP by 0.5% in 1-shot classi-

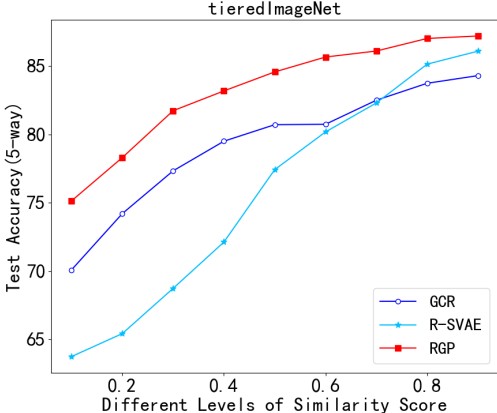

Figure 3: 5-way performance with different levels of similarity score for base and novel classes on tieredImageNet.

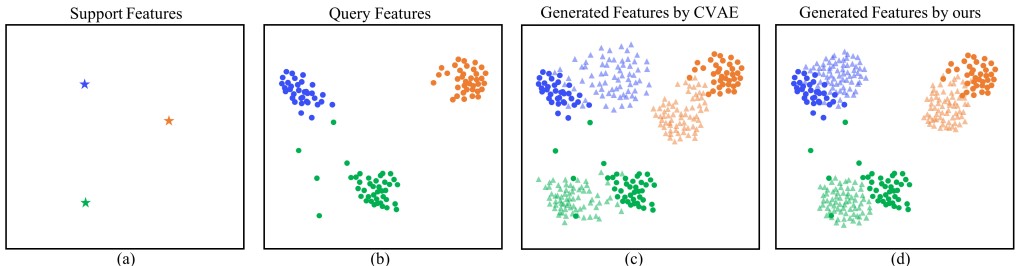

Figure 4: **Feature Visualization.**We visualize the features of the original and generated data. The transparent points are the generated features and the non-transparent points are the original features. (a): the support set features. (b): the query set features. (c): the generated features by CVAE. (d): the generated features by ours.

fication on miniImageNet. To verify the effect of non-representative data, we compared RGP w/o and RGP without non-representative data. Experimental results show that RGP slightly outperforms RGP w/o. Another interesting finding is that 5-shot is not as good as 1-shot for classification. RGP outperforms previous state-of-the-art methods by $1.18 \sim 3.34\%$ for 1-shot and $0.2 \sim 1.14\%$ for 5-shot.

Since our method trains all classes together, we don't mind whether the base class distribution differs from the novel class distribution. However, in order to verify that RGP has better performance on tasks of novel classes that differ greatly from the base class, we re-partition the dataset. According to the similarity size (between 0-1) of the prototype of the base class and the novel class, we divided the dataset into 9 levels. As shown in Figure 3, the performance of RGP is better when there are large differences between the base class and the novel class than R-SVAE Xu & Le (2022)and GCR Li et al. (2019). Moreover, as the similarity of the curve increases, it becomes more stable, indicating that RGP is less affected by the distribution of data.

## 4.4 VISUALIZATION

During training, Figure 4 illustrates the process of generating data with different methods using tieredImageNet datasets. Each of the four images in Figure 4 depicts the features of three novel classes, which are mapped to 2-d space using t-SNE Van der Maaten & Hinton (2008). Figure 4 (d) illustrates the rationality and effectiveness of our synthesis strategy since the features generated by our method are closer to the real features than CVAE alone as shown in Figure 4 (c).

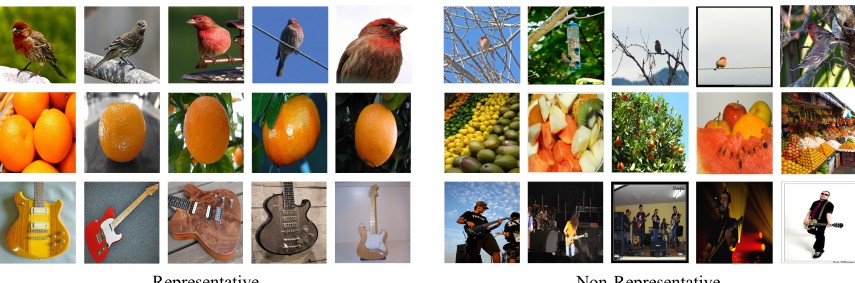

Representative                    Non-Representative

Figure 5: Some examples for representative samples (left) and non-representative samples (right) from miniImageNet.

Table 2: 1-shot classification accuracies (%) with different generation methods on miniImageNet and tieredImageNet.

| model | miniImageNet | tieredImageNet |
|---|---|---|
| VAE | 73.62±0.41 | 75.64±0.37 |
| CVAE | 75.18±0.64 | 78.38±0.25 |
| RGP(NSS) | 74.13±0.23 | 77.24±0.16 |
| RGP | **76.42±0.35** | **80.35±0.47** |

According to different selection criteria, Figure 5 illustrates representative and non-representative samples. The representative samples are closer to the center of the distribution and therefore have a higher probability and contain the main clean features that make them easier to identify. Samples selected at the tail of the distribution have a lower probability of being assigned and may contain features relating to other categories and therefore are misclassified or hard to identify.

### 4.5 ABLATION STUDY

Our main experiments involve synthesizing samples in four parts. We compare synthesis strategies that use only one step separately to our approach in the ablation experiments: VAE generation with Wang et al. (2018), CVAE generation with semantic embedding as a condition Xu & Le (2022) and no synthesis strategies method RGP(NSS).

As shown in Table 2, the 1-shot performance in different sample generation strategies on miniImageNet and tieredImageNet datasets. We can see that the VAE-based approach fixes the parameters obtained by training the base class, which is less effective than using CVAE that span different distributions in terms of semantic embeddings. The RGP(NSS) only use the generate features after CVAE but without data of novel classes. Finally, the classification performance of both is inferior to our proposed approach which does not rely on the assumption of similar distributions.

## 5 CONCLUSION

A new few-shot learning with representative global prototype framework is proposed to alleviate the problem of poor generalization ability. In order to obtain a representative global prototype with strong generalization ability on the performance of the novel class, a joint training strategy is proposed, which trains both the base class and the novel class on representative and non-representative data. Second, to alleviate the problem of sample imbalance in joint training, a sample synthesis strategy that combines base classes and novel classes conditioned on semantic embeddings is proposed, which is not restricted to the strong assumption of meta-learning. As a final step, three loss functions are combined to train the entire model. Experimental results show that our method outperforms existing few-shot methods. In the future, we will explore the impact of different sample selection methods on model classification performance.

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
