# OpenReview forum: "Few-Shot Learning with Representative Global Prototype"
_ICLR.cc/2023/Conference — Submitted to ICLR 2023_

### Official Review · Reviewer_5Wqr · 2022-10-22

**Confidence:** 2
**Correctness:** 3
**Technical Novelty And Significance:** 3
**Empirical Novelty And Significance:** 3
**Recommendation:** 5

**Clarity, Quality, Novelty And Reproducibility:**

The paper is well-written and easy to follow. It considers an interesting problem in few-shot learning, and has originality. But more detailed disscussions experiments should be added. It provides detailed experimental implementation. But the proposed method contains many modules which is not easy to reproduce.

**Strength And Weaknesses:**

Strength:

1.	The paper considers an interesting problem in the few-shot learning task: the data distribution of novel classes and base classes can be different. The different distributions lead to the performance degradation of few-shot learning methods. Correspondingly, the paper proposes to generate more samples for novel classes, which is reasonable.

2.	The paper is well-written and easy to follow.

3.	The proposed method achieves the state-of-the-art performance on miniImageNet and tieredImageNet datasets.

Weakness:

1.	The paper claims that the data distribution of novel classes and base classes could be different, resulting in performance degradation. It seems that the proposed method is more suitable for cross-domain few-shot classification. On miniImageNet and tieredImageNet datasets, the base classes and novel classes follow a similar distribution, which can not show the effectiveness of the proposed method. The experiments on cross-domain few-shot classification tasks should be discussed and added.

2.	More detailed discussions and experiments should be added. The paper should consider dividing novel tasks into multiple different levels based on the similarity with based classes, and verify whether the proposed method is more effective on the tasks that are more different from based classes.

3.	In the experimental setting, the proposed method uses CLIP to extract the semantic embedding for generating new novel features. CLIP contains external information. So it is unfair to compare the proposed method  with other methods. For a fair comparison, it is better to replace CLIP with another way.


**Summary Of The Paper:**

This paper proposes a few-shot learning with representative global prototype method. It uses selected representative and non-representative samples to optimize the representative global prototypes. It also utilizes semantic embedding to generate new samples of novel classes in order to alleviate the imbalance problem. The experimental results show that the proposed method can improve few-shot learning tasks and achieves state-of-the-art performance.

**Summary Of The Review:**

The paper considers an interesting problem in few-shot learning, but more detailed disscussions experiments should be added.

---

### Official Review · Reviewer_HxLp · 2022-10-23

**Confidence:** 4
**Correctness:** 2
**Technical Novelty And Significance:** 2
**Empirical Novelty And Significance:** 2
**Recommendation:** 3

**Clarity, Quality, Novelty And Reproducibility:**

Novelty

The proposed method combines the idea of the global prototype (Li et al., 2019) and representative samples (Xu et al., 2022). The global prototype (Li et al. 2019) uses the similarity between input and global representation, but the method is built on a meta-learning framework. (Xu et al., 2022) proposed the selection of representative samples and the CVAE sample generation model. The authors introduced the modules of (Xu et al., 2022) into the global representation (Li et al., 2019) but without meta-learning. The method seems to be good, but the components are not novel.

A novel point might be the introduction of non-representative samples. But this is not highlighted.

Quality

The ablation study is insufficient.
One contribution is the joint training with representative and non-representative samples. The effects of introducing the non-representative samples are not shown.
 Also the second contribution is to propose a sample synthesis strategy ( but this is the same as Xu et al., (2022)). The effect without using the sample synthesis is not evaluated.

Clarity

P.1 The “metric novel class data” is unclear. Also, “Metric presentative sample”(P2), “metric the global prototype” (P4), “metric the new global prototype”(P4) are not clear.

P1. The third paragraph seems to describe a specific paper. The author should clarify the paper.

P2. “Metric representative sample and representative global prototype to generate a global loss” is not a sentence.

P3. “the base and novel class training the representative and non-representative samples”   is a typo

P4. “to form a support set to form a support set”  is a typo

P4. S={$x_j$ | p($x_j$) | $\mu_i$, $\sigma_i$ > $\epsilon$ } is a typo.









**Strength And Weaknesses:**

Strengths
- The proposed approach, which uses based and novel classes to train and generate the samples for novel classes, seems reasonable.
- The proposed method shows better performance than the state-of-the-art methods.

Weaknesses
- Components of the proposed method are not novel (see Novelty below).
- The ablation study is insufficient (see Quality below )
- The writing of this paper needs improvement. (see Clarity below)


**Summary Of The Paper:**

This paper proposes a joint training strategy for few-shot learning via representative and non-representative samples to enhance the generalization to novel classes.
The representative samples are used for the global prototype, and the non-representative samples are further used for classification loss.
A sample generation method is also used to generate samples of novel classes.
The proposed method exhibits state-of-the-art performance on the miniImageNet and tieredImageNet datasets.


**Summary Of The Review:**

Although the proposed method is reasonable and achieves state-of-the-art performance, the proposed method only combines existing components, and this paper does not have novel aspects for few-shot learning. The ablation study is insufficient, and the writing needs improvements.

---

### Official Review · Reviewer_pA9r · 2022-10-24

**Confidence:** 2
**Correctness:** 2
**Technical Novelty And Significance:** 2
**Empirical Novelty And Significance:** 2
**Recommendation:** 5

**Clarity, Quality, Novelty And Reproducibility:**

The paper writing is not clear enough.
Quality\Novelty\Reproducibility: It woud be helpful if the authors can address my above-mentioned doubts.



**Strength And Weaknesses:**

strength: This work proposes a few-shot learning with representative global prototype method.

weakness: I have some questions, described below. As stated in 3.1, there are a total of N classes of samples and the support set also has N classes. To build the support set, the authors stated that they select a representative set of training samples of the ''base class''. Do they mean base classs and novel classses? Besides, the authors select the sample using p, mean vector of class i and covariance of class i. Are the mean vector and covariance of class i being fixed for using all samples within this class i? I am confused about "which is its covariance Σi is small if Σi  is fixed for a given class i". To generate samples for novel classes, the authors combine the base class data and semantic embedding ai. How can we obtain the semantic embedding, such as one-hot vector? Could the authors provide more explaination for  semantic embedding and how the samples for novel classes are generated? More importantly, how many N way K shot tasks are used to evaluate the performance? Here, N means subset of all novel classes rather than total classes. The proposed RGP is used to train a standard N way K shot or train their mentioned support set. I think the latter includes more noval task information than former. Therefore, it may be unfair to compare standard few-shot learning methods if the authors using all novel tasks. Besides, how about the computational cost at training and testing stage? Lastly, the authors mentioned their proposed RGP can break the barrier: the sample distribution of the novel class and the base class are similar. It would be better if the authors can provide some experimental results to support this argument.



**Summary Of The Paper:**

This submission introduces a new few-shot learning with representative global prototype framework. For one thing, they propose a novel jointly training strategy for few-shot learning via representative and non-representative samples. For another thing, they propose a sample generation strategy to generate samples of the novel class to solve the sample imbalance problem caused by limited data (few shots).



**Summary Of The Review:**

Somewhat novel and reasonable if the method is fair when comparing with its baselines.

---

### Decision · Program_Chairs · 2023-01-20

**Decision:**

Reject

**Justification For Why Not Higher Score:**

The submission is below the acceptance bar due to writing clarity and reproducibility concerns raised by the reviewers. The authors have not responded to the reviews.

**Justification For Why Not Lower Score:**

N/A

**Metareview: Summary, Strengths And Weaknesses:**

The submission proposes to tackle few-shot classification by training jointly on base and novel classes to learn a global representative prototype for each class. The prototype is computed by averaging the embeddings of representative examples of a given class, where "representative" is defined as having a high probability mass according to a Gaussian model of the class-wise embedding distribution. Non-representative examples are used to compute the classification loss. The submission also proposes to generate new samples for novel classes to alleviate the class imbalance problem. Results are presented on mini-ImageNet and tiered-ImageNet, where the proposed approach is claimed to achieve state-of-the-art performance.

Reviewers noted the proposed approach's strong performance, but expressed concerns about writing clarity and reproducibility (due to the many moving parts). The authors have not responded to the reviews.

I recommend rejection.

**Summary Of Ac-Reviewer Meeting:**

N/A